# Investigations on the Diesel Spray Characteristic and Tip Penetration Model of Multi-Hole Injector with Micro-Hole under Ultra-High Injection Pressure

Chang Zhai [1], Feixiang Chang [2], Yu Jin [3] and Hongliang Luo [4,*]

1  Renewable Energy Research Center, National Institute of Advanced Industrial Science and Technology (AIST), 1-2-1 Namiki, Tsukuba 305-8564, Ibaraki, Japan; chang.zhai@aist.go.jp
2  School of Mechanical Engineering, Jiangsu University of Science and Technology, Zhenjiang 212100, China; changfx@just.edu.cn
3  Institute for Energy Research, Jiangsu University, Zhenjiang 212013, China; y.jin@ujs.edu.cn
4  College of Power and Energy Engineering, Harbin Engineering University, Harbin 150001, China
*  Correspondence: luohl@hrbeu.edu.cn

**Abstract:** Increasing the injection pressure has a significant impact on atomization and combustion characteristics. Spray tip penetration serves as a vital parameter for fuel injection control and engine structure design. However, a reliable spray tip penetration model for ultra-high-pressure injection is currently lacking. To address this gap, this study establishes a theoretical 0-dimensional model for spray tip penetration under ultra-high pressure (300 MPa) conditions. The model is based on the conservation of momentum and phenomenological models. The new model divides spray tip penetration into two stages: Pre-breakup and post-breakup, with fuel injection rate and spray cone angle used as model inputs. To validate the model, high-speed camera observations and constant-volume chamber experiments are conducted to investigate the spray characteristics. The results indicate that the new spray tip penetration model demonstrates improved predictive accuracy across all experimental conditions.

**Keywords:** diesel fuel; spray penetration model; ultra-high injection pressure; 0-dimensional model

## 1. Introduction

Sustainability is a critical global issue (renewable energy, environmental conservation, reducing carbon emissions, etc.), and various research efforts are being made to address it [1–3]. Their work aims to develop innovative technologies, improve resource efficiency, and promote sustainable practices. Diesel engines are extensively utilized in various industries, including agriculture, construction, and shipbuilding, due to their significant torque and excellent economic performance [4–7]. However, in recent years, the engine industry has encountered significant challenges due to the continuous enhancement of emissions and efficiency [8–10]. To address these challenges, numerous advanced technologies, such as homogeneous charge compression ignition (HCCI) and low-temperature combustion (LTC), have been implemented to improve combustion performance and achieve cleaner combustion in diesel engines. The implementation of these technologies has led to a continuous increase in the injection pressure of diesel engines [11,12]. Currently, diesel engines operate with injection pressures surpassing 250 MPa. Meanwhile, previous studies on spray have shown that for direct-injection diesel engines, combustion and pollutant formation are largely influenced by the dynamics of spray and air mixing. Higher injection pressures directly affect diesel atomization and mixing performance [13,14]. Moreover, under high-load conditions, faster spray penetration contributes to improved air utilization and combustion rates [15]. Therefore, increasing injection pressure and reducing nozzle diameter are effective methods to enhance fuel atomization and accelerate mixing in diesel engines.

The 0-Dimensional (0-D) model is extensively employed in predicting spray and combustion characteristics due to its cost-effectiveness, minimal input parameters, and time efficiency [16,17]. Accurately predicting spray tip penetration is crucial for engine structural design and fuel injection strategy. Over the years, several models have been developed to estimate spray tip penetration, including the Wakuri model, the Hiroyasu model (which divides the injection process into two stages), and the Naber and Siebers model, among others [18–23]. However, these models have limited applicability in terms of the pressure range. In recent years, research on spray penetration models has continued [24–28]. Researchers, such as Desantes and Payri, have incorporated spray momentum as an input to the model to predict spray penetration [25,26]. The predicted spray tip penetration from these models has been compared with experimental results, demonstrating their ability to predict penetration, even during initial time intervals. Kostas et al. focused on the initial stage of spray tip penetration and observed that it follows a functional relationship of the form S(t) = At$^{(3/2)}$ until the tip velocity reaches its maximum value, where A is a constant and t is the time. This behavior is attributed to the influence of supersonic conditions on the initial stage of the spray [27]. Zhou and Li studied the spray tip penetration distance model throughout the injection process, dividing it into five stages. The calculated results from their newly developed models showed excellent agreement with experimental data [28]. Jia et al. observed that under ultra-high injection pressure conditions, the spray head produces two shock waves. Additionally, a comparison with predictive models revealed that shock waves influence spray tip penetration [29]. Table 1 summarizes the main spray tip penetration models established by researchers in different periods. From the table, it is evident that although numerous researchers have made significant progress in spray tip penetration models, most of these prediction models are applicable only to a relatively low-pressure range. Models for high injection pressure to ultra-high injection pressure scenarios are still lacking. Therefore, through the utilization of the momentum conservation theorem and a phenomenological approach, a mathematical expression between spray velocity and penetration is established. The phenomenological model was used as a base to divide the spray into two stages. Meanwhile, based on the spray tip penetration discussed in the first paragraph, the second paragraph considers the tip penetration in the second stage as an extension of the growth pattern observed in the first stage, using a momentum model for analysis. In other words, the results of the model in the first paragraph will directly affect the results of the second paragraph. Finally, the validity of this model is then verified by comparing it with experimental results under ultra-high injection pressure conditions.

**Table 1.** Spray tip penetration models.

| Model | Correlation | References | $P_{inj}$ Range |
|---|---|---|---|
| Wakuri | $S = 1.18 C_a^{0.25} \left( \frac{\Delta P}{\rho_a} \right)^{0.25} \left( \frac{Dt}{\tan(\theta)} \right)^{0.5}$ | Wakuri et al., 1960 [18] | 400–750 atm |
| Dent | $S = 3.07 \left( \frac{\Delta P}{\rho_a} \right)^{0.25} \left( \frac{294}{T_a} \right)^{0.5} (Dt)^{0.5}$ | Dent., 1971 [19] | 10–66 MPa |
| Hiroyasu and Arai | $S = 0.39 \left( \frac{2\Delta P}{\rho_l} \right)^{0.5} t \quad 0 < t < t_b$ <br> $S = 2.95 \left( \frac{\Delta P}{\rho_a} \right)^{0.25} (Dt)^{0.5} \quad t > t_b$ <br> $t_b = 28.65 \frac{D\rho_l}{\sqrt{\Delta P \rho_a}}$ | Hiroyasu and Arai., 1990 [20] | 7–150 MPa |
| Schihl | $S = 1.414 C_v^{0.5} \left( \frac{\Delta P}{\rho_a} \right)^{0.25} \left( \frac{Dt}{tan(\theta)} \right)^{0.5}$ | Schihl et al., 1996 [21] | 7–160 MPa |
| Naber and Sibers | $\widetilde{S} = \left[ \left( \frac{1}{\widetilde{t}} \right)^n + \left( \frac{1}{\widetilde{t}^{0.5}} \right)^n \right]^{-\frac{1}{n}}$ | Naber and Sibers., 1999 [22] | 75–160 MPa |

**Table 1.** *Cont.*

| Model | Correlation | References | $P_{inj}$ Range |
|---|---|---|---|
| Arrègle | $S = D^{0.307} P_{inj}^{0.262} \rho_a^{-0.406} t^{0.568}$ | Arrègle et al., 1999 [23] | 30–110 MPa |
| Sazhin | $S = v_{d0}t - 0.5 v_D^2 \gamma K v_D^{1.5} t^{2.5}$ | Sazhin et al., 2001 [24] | 90 MPa |
| Desantes and Payri | $S = 1.26 \cdot \rho_a^{-0.25} M_i^{0.25} (t)^{0.5} \left(\tan\left(\frac{\theta}{2}\right)\right)^{-0.5}$ | Desantes and Payri et al., 2006 [25] | 50–130 MPa |
| Desantes and Payri | $S_1 = k_p \left(\tan\frac{\theta}{2}\right)^{-0.5} \rho_a^{-0.25} M_0^{0.25} t^{0.5}$ <br> $S_2 = k_p \left(\tan\frac{\theta}{2}\right)^{-0.5} \rho_g^{-0.25} M_0^{0.25} (t - \Delta t + \varphi)^{0.5}$ | Desantes and Payri et al., 2006 [26] | 30–130 MPa |
| Kostas | $S = A(t)^{1.5}$ <br> Hiroyasu's model after the intersection | Kostas et al., 2009 [27] | 50–100 MPa |
| Xinyi Zhou and Tie LI | $S = K_1 \left(\frac{2\Delta P}{t_p}\right)^{1/2} \rho_l^{-1/2} t^{3/2} \quad 0 < t \le t_b$ <br> $S = K_2 \left(\frac{\Delta P}{t_p}\right)^{1/4} \rho_a^{-1/4} (D)^{1/2} t^{3/4} \quad t_b \le t \le t_p$ <br> $S = K_2 \left(\frac{\Delta P}{\rho_a}\right)^{1/4} (D)^{1/2} (t)^{1/2} \quad t_p \le t \le 2t_i$ <br> $S = 2^{1/2} K_2 \left(\frac{\Delta P}{\rho_a}\right)^{1/4} (D)^{1/2} (t_i)^{1/4} (t - t_i)^{1/4} \quad t \ge 2t_i$ | Tie LI., 2021 [28] | 90–150 MPa |

## 2. Experimental System and Conditions

*Experimental Setup*

In this study, we employed three Denso ten-hole electromagnetic injectors for the experiment. These injectors had different hole diameters, specifically 0.07, 0.101, and 0.133 mm. One of the injectors had a 0.07 mm diameter, specially manufactured for experimental purposes, while the other two injectors were mass-produced. The length of the injector nozzle was 0.8 mm, and the angle between the axis of each injector hole and the central axis of the injector was 77.5°. The specific structure and parameters of the injector can be observed in Figure 1 and Table 2, respectively.

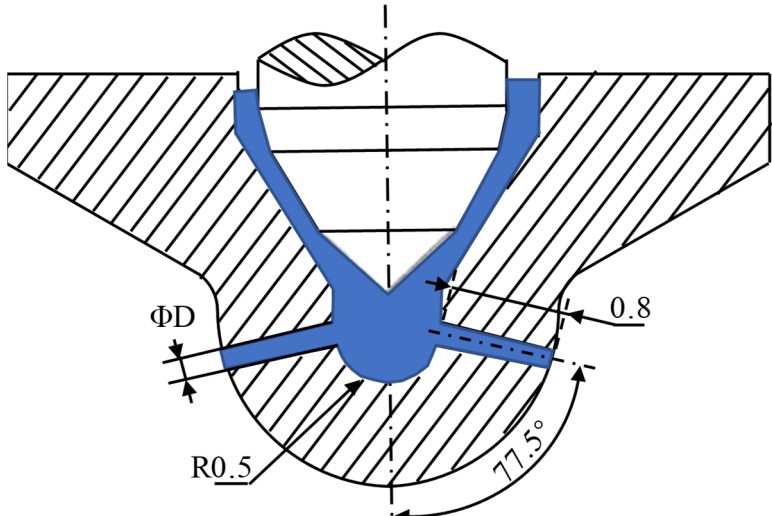

**Figure 1.** Schematics of 10-hole nozzles.

**Table 2.** Injector parameters.

| Items | Value |
|-------|-------|
| Injectors | Denso G4S (solenoid injector) |
| Type | Mini-Sac |
| Hole number | 10 |
| Umbrella angle [°] | 155 |
| Nozzle-hole diameter (D) [mm] [mm] | 0.07 0.101 0.133 |
| Hole length [mm] | 0.8 |
| Sac radius [mm] | 0.5 |

Figure 2 shows the injection system and observation system used in this study. The diffuser background illumination method (DBI) was used to obtain the characteristics of the spray. The light source of the observation system was provided by the high brightness LED light source (by Altec company). The light made by the LED light source streamed into the constant volume combustion chamber (CVCC) through the diffuser and was captured by the high-speed camera after passing the target spray. The delay generator (DG645) simultaneously controlled the ECU (Electronic Control Unit) of the injector and the high-speed camera to ensure that the high-speed camera could smoothly capture the target spray. The injection pressure of the experiment was provided by the high-pressure Common Rail system, which could provide a stable injection pressure, and the maximum pressure could reach more than 300 MPa. The nitrogen cylinder was connected to the constant volume combustion chamber through the pressure reducing valve and switch valve. The constant volume combustion chamber was connected with the pressure gauge to ensure pressure control in the chamber. Table 3 shows the optical system configurations in the DBI experiment. A visible lens (Nikon, 105 mm, f/4.8) coupled with a NAC-MEMRECAM HX-3 high-speed camera was used. They provide frame rates of up to 20,000 frames per second. It should be noted that the injector was embedded over the chamber at an angle of approximately 37.5° from the horizontal direction. Therefore, the target spray was inclined at an angle of 25° with the vertical instead of vertically downward. Thus, the results of the multi-hole injectors shown in the figure were vertically scaled by multiplying it by a factor of 1/cos25°.

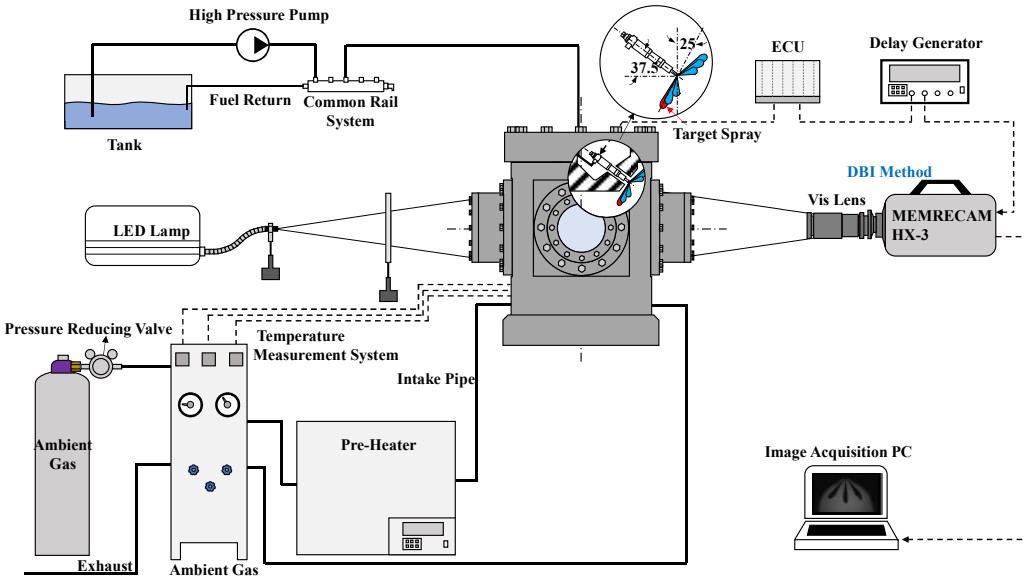

**Figure 2.** Experimental setup.

**Table 3.** Optical system configurations.

| Items | Value |
|---|---|
| High-speed camera | NAC-MEMRECAM HX-3 |
| Lens | Nikon, 105 mm |
| Light source | Altec LED lamp |
| Pulse generator | DG535 |
| Resolution | 640 × 640 |
| Exposure [ms] | 0.005 |
| Framerate [fps] | 20,000 |
| Aperture sizes [1/f] | 4.8 |

The experimental conditions are shown in Table 4. The JIS#2 diesel was used as the test fuel in the experiment. The main properties of the fuel are shown in Table 5. In this study, we used the same injection duration (2.3 ms). The injection pressure was from 100 to 300 MPa. The ambient temperature of the experiment was 300 K. Nitrogen with stable chemical properties was used as the ambient gas to fill the whole constant volume combustion chamber. According to the state equation of an ideal gas, the ambient pressure was calculated as 0.88, 1.32, and 1.76 MPa.

**Table 4.** Experimental conditions.

| Injection Condition | |
|---|---|
| Fuel | Diesel (JIS#2) |
| Injection Duration [ms] | 2.3 |
| Injection Pressure ($P_{inj}$) [MPa] | 100 200 300 |
| Nozzle Hole Diameter (D) [mm] | 0.07 0.101 0.133 |
| Injection Amount [mg] | 28.2~157.89 (Depends on $P_{inj}$ and D) |
| **Ambient Condition** | |
| Ambient gas | Nitrogen |
| Gas Density ($\rho_{amb}$) [kg/m$^3$] | 10/15/20 |
| Ambient Temperature [K] | 300 |
| Ambient Pressure [MPa] | 0.88 1.32 1.76 |

**Table 5.** Main properties of the fuel.

| Fuel Property | Diesel (JIS#2) |
|---|---|
| Density @ 15 °C [kg/m$^3$] | <860 |
| Kin.Viscosity @ 30 °C [mm$^2$/s] | >2.5 |
| Flash point [°C] | >60 |
| Flow point [°C] | <−7.5 |
| Cetane number | >45 |
| Ignition point [°C] | >50 |
| Oxygen content [wt%] | <1 |

## 3. Results and Discussion

### 3.1. Injection Rate and Spray

The fuel injection rate is a crucial parameter that directly impacts the atomization process, influencing the quality and characteristics of the spray. Several methods exist for measuring the fuel injection rate, including the Bosch long tube method, Zeuch method, momentum flux measurement method, etc. [30–32]. For this study, considering safety and stability, we opted for the Bosch long tube method as the measurement technique for fuel injection rate. Figure 3 illustrates the variations in injection rate under different hole diameters and injection pressure conditions. It is evident from the figure that the injection rate increases significantly with higher injection pressure and larger hole diameter. This

can be attributed to the fact that the injection rate is primarily governed by the effective flow area and injection speed. The injection rate can be divided into three main stages. In the initial stage, the injection rate rises rapidly. When the injection pressure is higher, the slope of the injection rate also increases, predominantly influenced by the pressure in the sac. During the stable stage, the injection rate gradually stabilizes, with the injection pressure having minimal effect on the duration of this stage. However, as the injector hole diameter increases, the stable stage duration decreases, primarily due to the movement of the injector needle and the sac pressure [33]. In the final stage, the injection rate decreases rapidly until the needle is closed. Similar to the initial phase, higher injection pressure leads to a faster decrease in the end-stage.

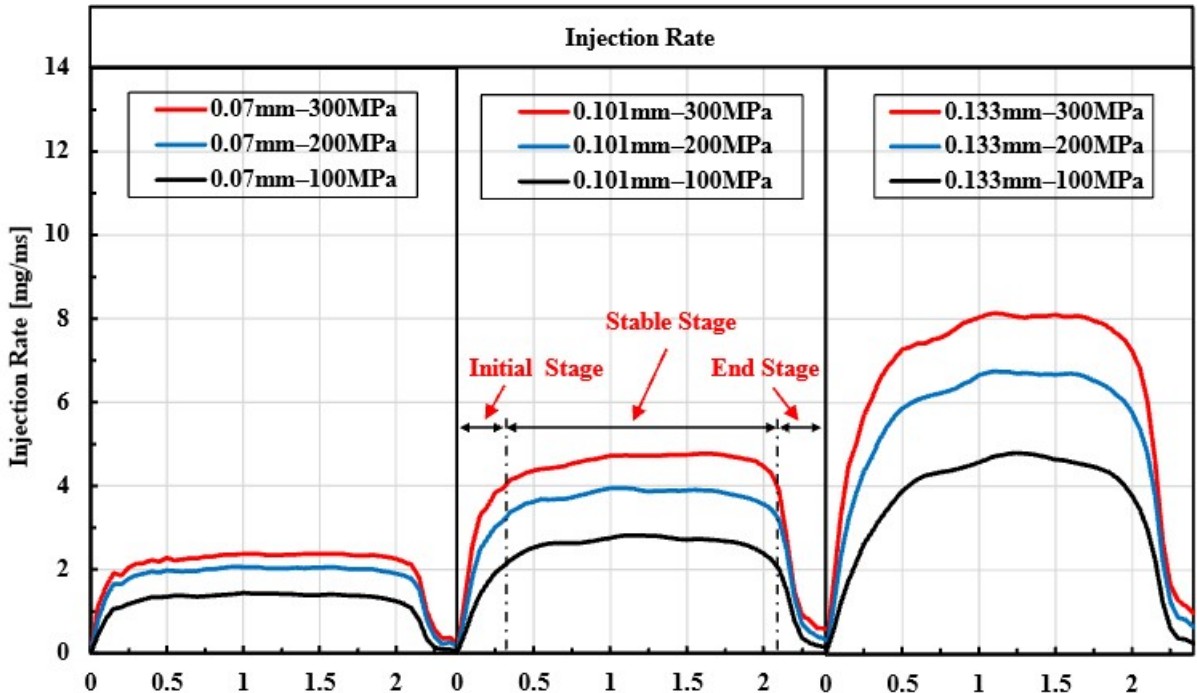

**Figure 3.** Injection rate with different hole diameters.

Figure 4 shows the non-evaporating spray images obtained through the DBI method. The figure clearly illustrates that under ultra-high injection pressure conditions, the spray boundary of the injector with a larger hole diameter becomes highly unstable. Particularly during the initial stage of injection, the spray shape experiences severe deformation. Additionally, it should be noted that the spray tip penetration of the target spray differs from that of the other sprays during the initial stage. However, as the spray develops, the disparity in tip penetration among the sprays gradually decreases, eventually leading to similar tip penetrations for all sprays.

Reducing the hole diameter and injection pressure effectively enhances the spray stability, as depicted in Figure 4c,d. This phenomenon can be attributed to the increased complexity of internal flow within the injector nozzle and intensified cavitation in the nozzle resulting from higher injection pressure and smaller hole diameter. These factors contribute to the observed spray instability.

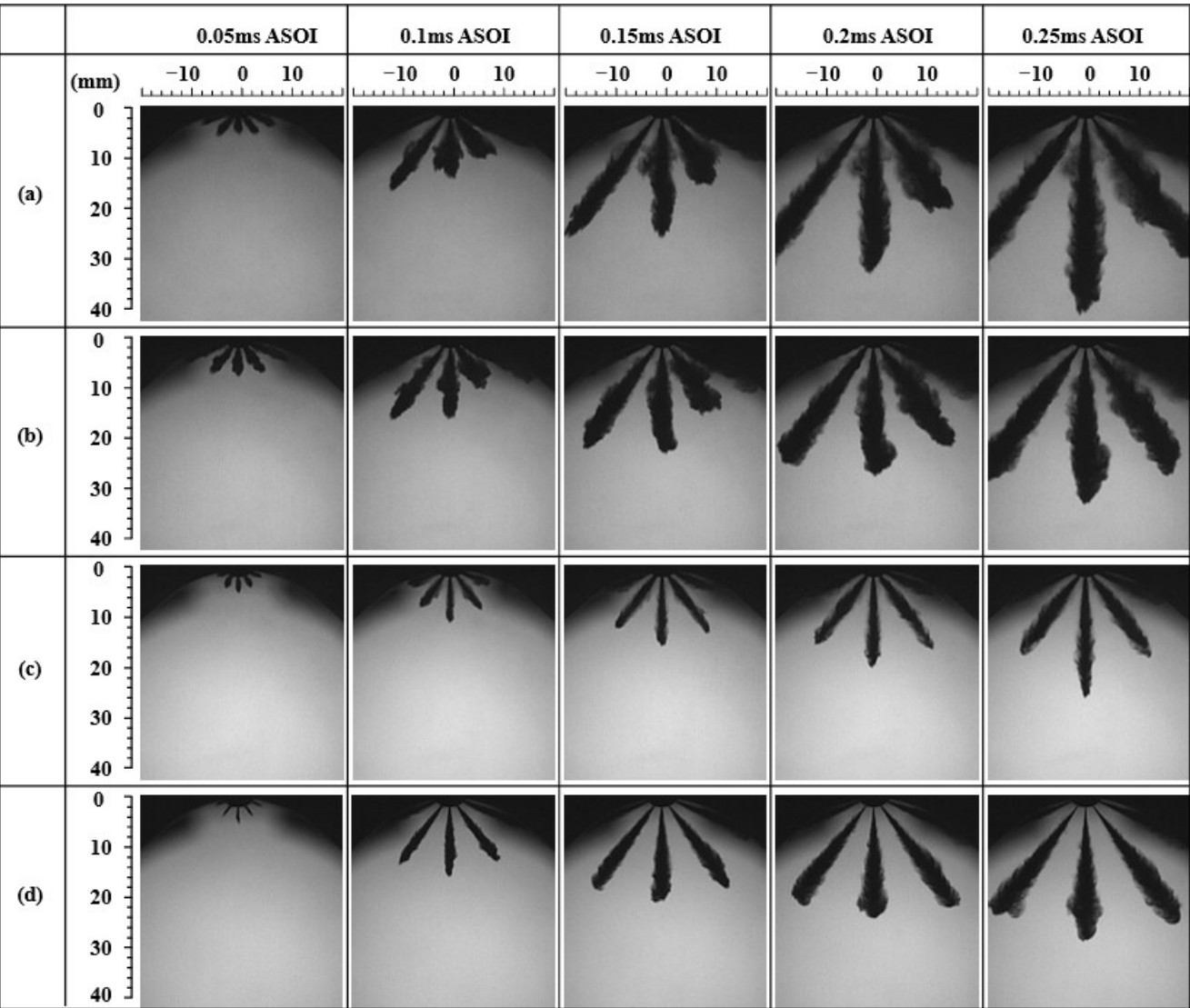

**Figure 4.** Diffuser background illumination images of spray. (**a**) $P_{inj}$ = 300 MPa, D = 0.133 mm, $\rho_a$ = 20 kg/m$^3$ (**b**) $P_{inj}$ = 300 MPa, D = 0.133 mm, $\rho_a$ = 10 kg/m$^{3.}$ (**c**) $P_{inj}$ = 100 MPa, D = 0.133 mm, $\rho_a$ = 20 kg/m$^3$ (**d**) $P_{inj}$ = 300 MPa, D = 0.07 mm, $\rho_a$ = 20 kg/m$^3$.

### 3.2. Image Proceeding and Macroscopic Characteristics of the Spray

In the preceding analysis, our focus was on examining the spray characteristics using non-evaporating spray images obtained through the DBI method. However, relying solely on non-evaporating spray images may not provide a comprehensive understanding of the details. To overcome this limitation, we employed MATLAB software to process the original non-evaporating spray image. The specific image processing steps and definitions are illustrated in Figure 5. Firstly, we subtracted the target spray image (Figure 5b) from the background image (Figure 5a). Secondly, the resulting image was subjected to binarization. For the binarization process, we selected 15% of the maximum value of the image as the threshold [34]. The binarized image is depicted in Figure 5c.

Finally, the binarized image is utilized to define the macro characteristics of the spray. In this context, the distance between the tip of the nozzle and the tip of the spray is selected as the spray tip penetration. By adopting this definition, we obtain more detailed macroscopic characteristics of the spray, which are presented in Figure 6.

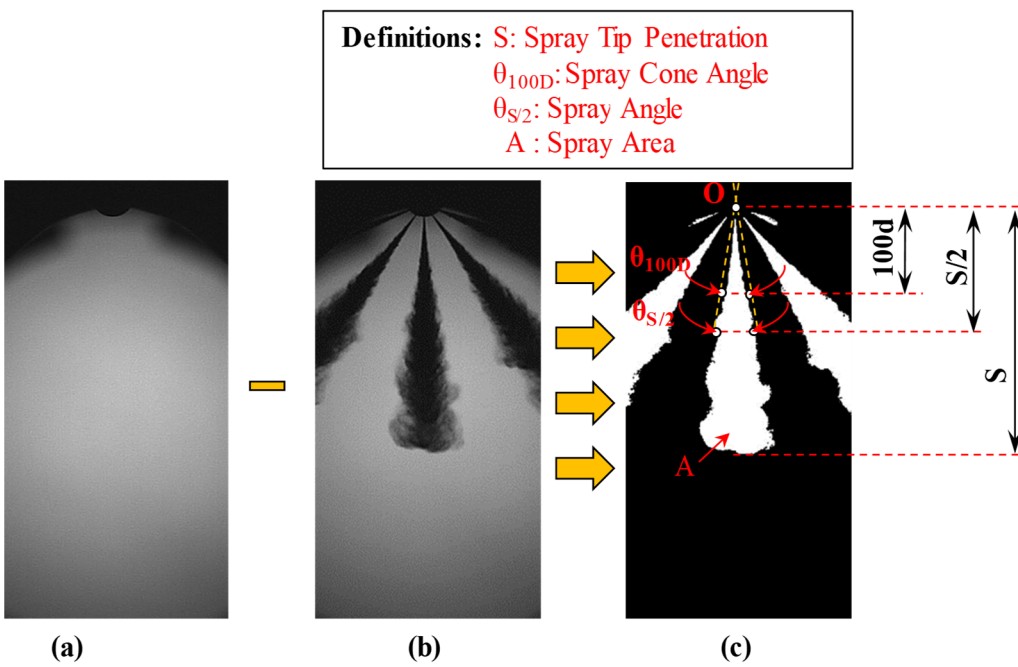

**Figure 5.** Definition of spray properties. (**a**) Background image (**b**) Spray image (**c**) Processed image.

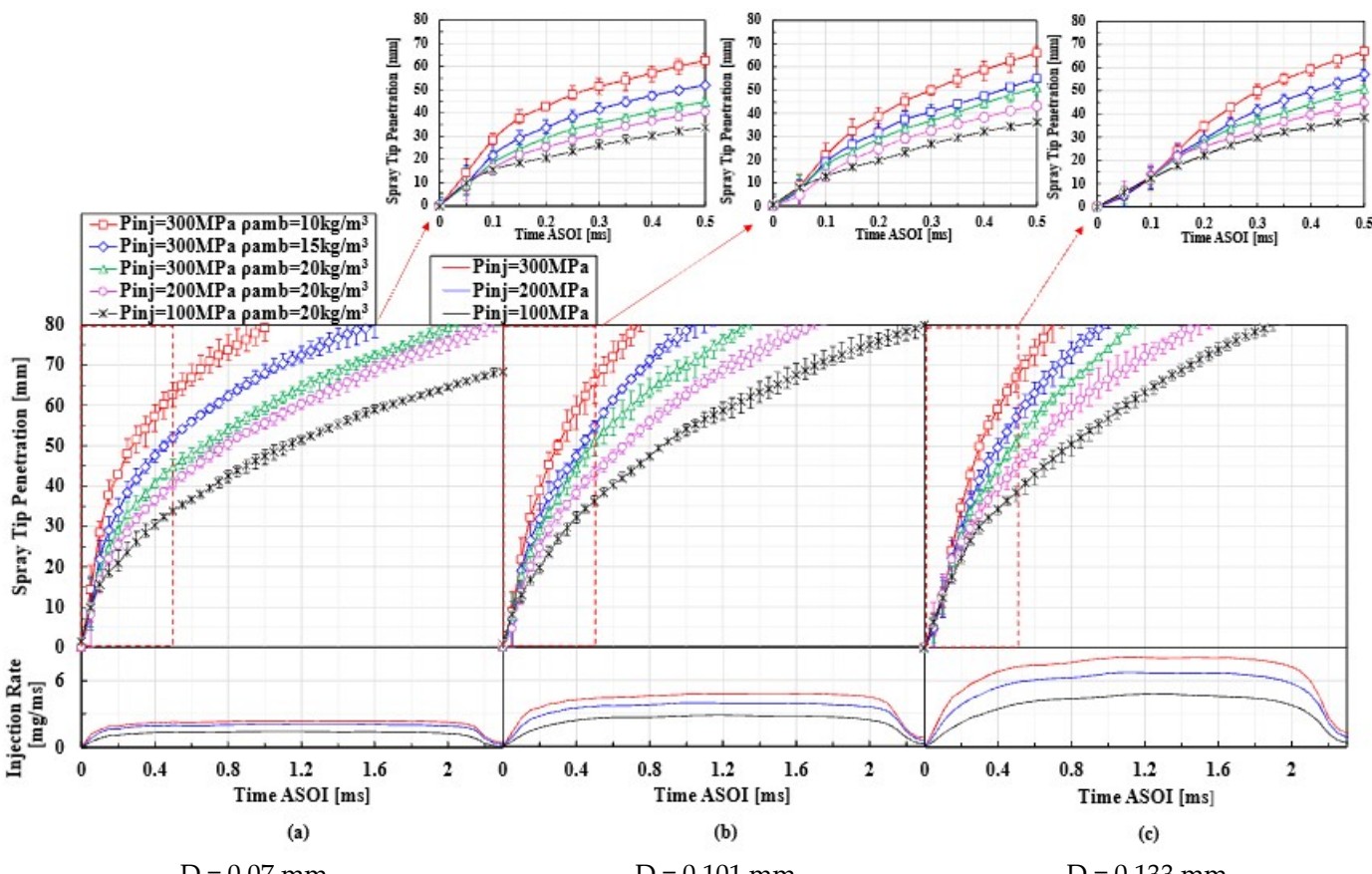

**Figure 6.** Spray tip penetration with different conditions.

From the figure, it is evident that the spray tip penetration increases with higher injection pressure, lower ambient density, and larger hole diameter. However, as the injection pressure increases from 200 to 300 MPa, the effect of injection pressure on the

spray tip penetration diminishes significantly. In contrast to the impact of injection pressure, the reduction in ambient density amplifies the effect on spray tip penetration (comparing the decrease from 20 to 15 kg/m$^3$, the decrease from 15 to 10 kg/m$^3$ has a greater impact on spray tip penetration). Additionally, it can be observed that the error bar of the spray tip penetration is larger when the ambient density is low, indicating that reducing the ambient density increases the instability of the spray tip penetration. Furthermore, when examining the spray tip penetration during the initial stage, it can be noted that the spray tip penetration is relatively similar under the same injection pressure. Comparing the results for different hole diameters, it can be observed that, during the initial stage, injectors with larger hole diameters exhibit longer spray tip penetration in a similar stage. This may be attributed to the larger mass flow and momentum of injectors with larger hole diameters. Moreover, it is evident that the spray tip penetration error bar is larger for injectors with larger hole diameters when comparing results for injectors with different hole diameters, indicating that increasing the hole diameter increases the instability of the spray.

*3.3. Theoretical Model Analysis*

The Hiroyasu model [20] defines the injection process of spray as two stages, before and after breakup.

$$S = 0.39 \left( \frac{2 \triangle P}{\rho_l} \right)^{0.5} t \qquad t < t_b \tag{1}$$

$$S = 2.95 \left( \frac{\triangle P}{\rho_a} \right)^{0.25} (Dt)^{0.5} \quad t \geq t_b \tag{2}$$

$$t_b = 28.65 \frac{D\rho_l}{\sqrt{\triangle P \rho_a}} \tag{3}$$

where $S$ is spray tip penetration, $\triangle P$ is difference between injection pressure and ambient pressures, $D$ is nozzle hole diameter, $\rho_a$ is gas density, $\rho_l$ is fuel density, $t_b$ is breaking time.

In this study, we still define the development stage of spray as two stages, one stage before crushing and the other being completely broken. The development state of spray in these two stages is different from the development rule, as shown in Figure 7.

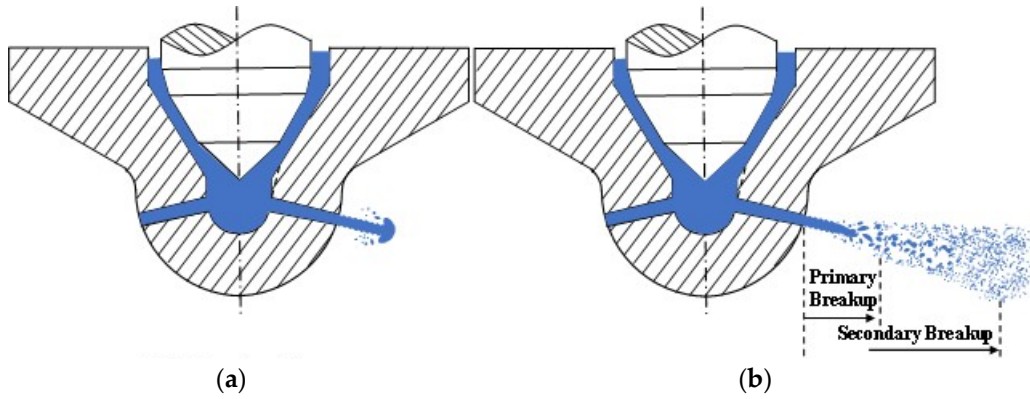

(**a**)          (**b**)

**Figure 7.** Schematic diagram of the breakup stage. (**a**) Before breakup, (**b**) After breakup.

## 4. Spray Tip Penetration Model

*4.1. Hypotheses*

The hypotheses assumed to carry out the theoretical derivation of the model are the following:

(a)   The gas in the environment remains non-flowing and the density and temperature remain constant.

(b)   The spray is an axisymmetric rotating body.

(c)  Before the breakup, the change of outlet velocity will directly affect the spray tip velocity.

(d)  After the breakup, the spray is assumed to be a gas jet, and the radial distribution of the spray velocity satisfies the following formula [25,35].

$$\frac{v}{v_m} = \left[ 1 - \left( \frac{r}{R} \right)^{1.5} \right]^2 \tag{4}$$

where $r$ is the radius position of any spray cross-section, $R$ is the maximum radius of the cross-section, $v$ is the speed of the spray cross-section where the radius is $r$, and $v_m$ is the speed at the center axis of the spray cross-section.

### 4.2. Before Breakup

At the initial stage of spray, near the nozzle exit, there is a very small spacing between broken droplets and ligaments, and the dynamic behavior of droplets will be greatly affected by adjacent droplets. Hiroyasu and others found that the penetration of diesel oil occurred within and outside the broken length at different rates. According to Kelvin–Helmholtz/Rayleigh–Taylor hybrid model [36], the crushing length is determined as follows:

$$L_b = U_r \tau_{KH} = \frac{1}{2} B_1 D \sqrt{\frac{\rho_l}{\rho_a}} \tag{5}$$

where $B_1$ is a constant, according to the research results of Beale and Reit, $B_1 = 40$. $U_r$ is relative drop/gas velocity. $\tau_{KH}$ is the breakup time obtained by the Kelvin–Helmholtz/Rayleigh–Taylor hybrid model.

It is assumed that the head speed delay caused by the change in exit velocity in the spray flow field is ignored before breaking. In other words, the change in exit velocity will directly affect the spray head, so the equation can be obtained.

$$L_b = \frac{1}{2} B_1 D \sqrt{\frac{\rho_l}{\rho_a}} = K_0 v_{\max} t_b \tag{6}$$

where $t_b$ is the breaking time, $v_{\max}$ is the maximum velocity of the injection, and the outlet velocity can be calculated by the measured injection rate.

Therefore, we can draw the model of the first penetration length of the spray:

$$S_1 = K_0 v_{\max} t_b \qquad (t < t_b) \tag{7}$$

### 4.3. Theoretical Model

Now, we construct a spray penetration model of the spray stable stage. Figure 8 shows a schematic diagram of the structure of the spray stable area. By using the intersection of two straight lines on the outer boundary of the spray as the origin "O" and establishing a coordinate system, according to the law of conservation of momentum, we know that the momentum of any cross-section remains constant. Taking the moment $t_{(s+1)}$ as an example, as shown in section A-A in Figure 8, Equation (8) can be obtained by integrating the section [25,26]:

$$\pi \rho_l r_0{}^2 v_0{}^2 = \int_0^R 2\pi \rho_a v_{(s+1)}{}^2 r\, dr \tag{8}$$

where $\rho_l$ is the fuel density, $\rho_a$ is the air density, $v_0$ is the nozzle outlet velocity during injection stable stage, $v_{(s+1)}$ is the velocity of the A-A cross-section, $R$ is the cross-sectional diameter, $dr$ is the radial differential unit.

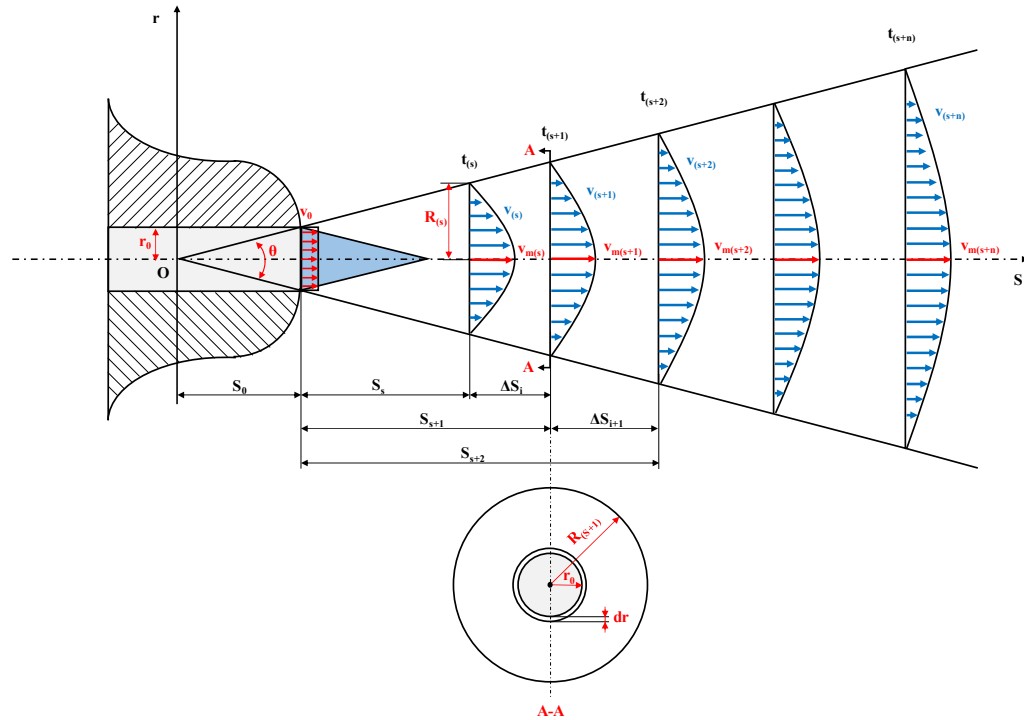

**Figure 8.** Schematic of Spray Steady Region.

$\delta$ parameter is introduced, and make:

$$\delta = \frac{r}{R_{(s+1)}} \tag{9}$$

and:

$$d\delta = \frac{dr}{R_{(s+1)}} \tag{10}$$

It can be seen from the assumption that the radial distribution of velocity satisfies Equation (4), substituting Equation (9) into Equation (4) to get:

$$\frac{v_{(s+1)}^2}{v_{m(s+1)}^2} = \left[\left(1 - \delta^{1.5}\right)^2\right]^2 \tag{11}$$

Equations (9)–(11) are substituted into Equation (8) to obtain:

$$\pi \rho_l r_0^2 v_0^2 = \int_0^1 R_{(s+1)}^2 2\pi \rho_a v_{m(s+1)}^2 \left[\left(1 - \delta^{1.5}\right)^2\right]^2 \delta d\delta \tag{12}$$

and so:

$$\frac{v_0}{v_{m(s+1)}} = \left(\frac{2\rho_a}{\rho_l}\right)^{0.5} \frac{R_{(s+1)}}{r_0} \left[\int_0^1 \left[\left(1 - \delta^{1.5}\right)^2\right]^2 \delta d\delta\right]^{-0.5} \tag{13}$$

can be calculated:

$$\int_0^1 \left[\left(1 - \delta^{1.5}\right)^2\right]^2 \delta d\delta \approx 0.0464 \tag{14}$$

and so:

$$v_{m(s+1)} = 3.28 \left(\frac{\rho_l}{\rho_a}\right)^{0.5} \frac{r_0 v_0}{R_{(s+1)}} \tag{15}$$

According to the spray structure in Figure 8, the following relationship can be obtained:

$$\frac{r_0}{R_{(s+1)}} = \frac{S_0}{S_0 + S_{(s+1)}} = \frac{1}{1 + \frac{S_{(s+1)}}{S_0}} = \frac{1}{1 + \frac{S_{(s+1)}\tan\left(\frac{\theta}{2}\right)}{r_0}} = \frac{r_0}{r_0 + S_{(s+1)}\tan\left(\frac{\theta}{2}\right)} \tag{16}$$

where $S_{(s+1)}$ is the distance from the cross-section to the nozzle hole, $S_0$ is the distance from the origin "O" to the nozzle outlet.

Equation (16) is substituted into Equation (15) to obtain:

$$v_{m(s+1)} = 3.28\left(\frac{\rho_l}{\rho_a}\right)^{0.5}\frac{r_0 v_0}{r_0 + \left(S_{(s+1)}\right)\tan\left(\frac{\theta}{2}\right)} \tag{17}$$

Now, we have got the relationship between the exit velocity $v_0$ and a certain position $v_{m(s+1)}$. Hiroyasu et al. [20] found that the penetration distance of the spray after breaking is proportional to $\sqrt{t}$, and based on this assumption, constructed the second equation of the Hiroyasu model. According to the results of Hiroyasu et al. [20] the second equation still passes through the breaking length $L_b$ at time $t_b$. The following equation is constructed:

$$L_b = \beta\sqrt{t_b} \tag{18}$$

We redefine the hypothesis here, assuming that the second section of the penetration distance model is gradually increasing based on the broken length, and the speed of spray tip is proportional to $\frac{1}{2\sqrt{(t-t_b+\gamma)}}$. Therefore, the following equation can be constructed:

$$S = L_b + \beta\sqrt{(t - t_b + \gamma)} - \beta\sqrt{\gamma} \tag{19}$$

Take the time derivative of Equation (19) to obtain Equation (20):

$$v_{tip} = \frac{dS}{dt} = \frac{\beta}{2\sqrt{(t - t_b + \gamma)}} \tag{20}$$

Let $t = t_b$, Can be calculated:

$$v_{L_b} = \frac{dS}{dt} = \frac{\beta}{2\sqrt{\gamma}} \tag{21}$$

We have constructed the spray velocity model at the stable stage, and now we estimate the spray motion. Obviously, the internal flow velocity of the spray at the center of the cross-section is different from the actual displacement velocity of the spray, but it directly affects the spray displacement velocity and is proportional to the spray displacement velocity. Therefore, we introduce the proportionality factor K:

$$v_{tip} = Kv_{m(s)} \tag{22}$$

and so:

$$\frac{\beta}{2\sqrt{\gamma}} = K3.28\left(\frac{\rho_l}{\rho_a}\right)^{0.5}\frac{r_0 v_0}{r_0 + (L_b)\tan\left(\frac{\theta}{2}\right)} \tag{23}$$

Let $\beta = K_1 6.56\frac{(\rho_l r_0 v_0)^{0.5}}{(\rho_a)^{0.25}}\tan\left(\frac{\theta}{2}\right)$ and so:

$$\gamma = K_2 \frac{\left(\frac{r_0}{\tan\left(\frac{\theta}{2}\right)} + L_b\right)^2}{r_0 v_0 (\rho_a)^{0.5}} \tag{24}$$

where $K = K_1 K_2$, and $K_1$ and $K_2$ are constants.

Therefore, we can get the model of the penetration after breakup as follows:

$$S = L_b + K_1 6.56 \frac{(\rho_l r_0 v_0)^{0.5}}{(\rho_a)^{0.25}} \tan\left(\frac{\theta}{2}\right) \left( \sqrt{(t - t_b + K_2 \frac{\left(\frac{r_0}{\tan\left(\frac{\theta}{2}\right)} + L_b\right)^2}{r_0 v_0 (\rho_a)^{0.5}})} - \sqrt{K_2 \frac{\left(\frac{r_0}{\tan\left(\frac{\theta}{2}\right)} + L_b\right)^2}{r_0 v_0 (\rho_a)^{0.5}}} \right) \tag{25}$$

In summary, we can get a complete penetration model:

$$S_1 = K_0 v_{max} t \qquad (t < t_b)$$
$$S = L_b + K_1 6.56 \frac{(\rho_l r_0 v_0)^{0.5}}{(\rho_a)^{0.25}} \tan\left(\frac{\theta}{2}\right) \left( \sqrt{(t - t_b + K_2 \frac{\left(\frac{r_0}{\tan\left(\frac{\theta}{2}\right)} + L_b\right)^2}{r_0 v_0 (\rho_a)^{0.5}})} - \sqrt{K_2 \frac{\left(\frac{r_0}{\tan\left(\frac{\theta}{2}\right)} + L_b\right)^2}{r_0 v_0 (\rho_a)^{0.5}}} \right) \qquad (t > t_b) \tag{26}$$

Breaking time $t_b$ can be obtained from Equation (6).

## 5. Model Validation

Refining the prediction of spray tip penetration is crucial for engine design and fuel injection. Numerous models have been developed by researchers to accurately estimate spray tip penetration. These models aim to capture the behavior of spray penetration and provide valuable insights for engine optimization. Among them, Hiroyasu's model and Dent's model are widely applied. In this study, to better compare the differences between different models, we compared our model with Hiroyasu's model and Dent's model. Additionally, to validate the accuracy of models, experimental results were compared with calculated results. Our model used the spray cone angle as an input parameter. Based on Hiroyasu's research, $K_0$ was set to 0.39. $K_1$ and $K_2$ were fixed values of 0.0162 and 0.0001, respectively. Figure 9 presents a comparison of experimental results and model predictions under different hole diameters, ambient pressures, and injection pressures. The blue dashed line represents the prediction of Hiroyasu's model, while the red dashed line represents the prediction of Dent's model. The black hollow circles represent the predictions of our model, and other colored markers represent the experimental results under different conditions.

From the figure, it can be observed that our predicted model aligns well with the experimental data under both ultra-high and high injection pressure conditions. However, both Hiroyasu's and Dent's models underestimate the spray tip penetration, with Dent's model performing slightly better than Hiroyasu's model. The discrepancy may arise from the different model coefficients, as the structure of the injector and the motion trajectory of the needle valve can affect the model coefficients. Furthermore, it is evident from the figure that as the hole diameter increases and the ambient pressure decreases, the error in our predicted model gradually increases, particularly in the initial stages of the model. This discrepancy may be attributed to the following factors:

1. The spray tip penetration exhibits cyclic variation, with slight differences in each injection, and there is still a certain level of measurement error in the experimental data.

2. This model used the spray angle as an input, and the accuracy of the model is directly affected by the magnitude and variation of the spray angle. Although we used measured spray angles as inputs, there is still an influence of error. Additionally, the spray angle is a function that changes with time, especially in the initial stages of the spray. In the future, it is necessary to consider varying spray angles as inputs for testing.

3. The accurate start of injection (SOI) also has a significant impact on the experimental results, directly affecting the time scale of the model. In this model, obtaining the SOI using high-speed video systems (HSV) introduces a large amount of error, resulting in a mismatch between the experimental and model results. In the future, it is recommended to redefine the SOI of the spray using microscopic imaging methods to obtain a more accurate SOI.

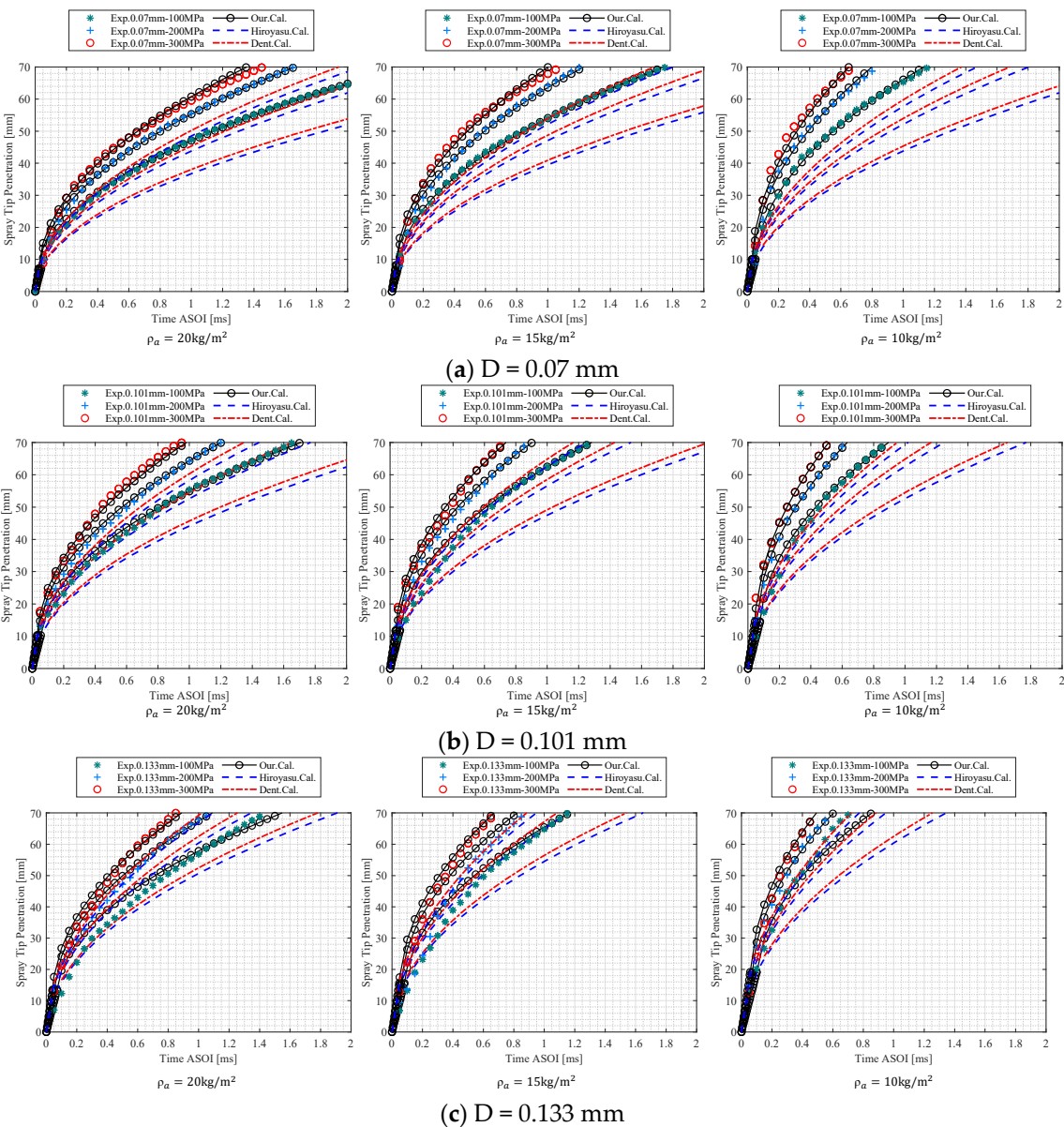

**Figure 9.** Comparison between models and experimental results with different conditions.

## 6. Conclusions

This study conducted visualization experiments on non-evaporative sprays with different diameters, injection pressures, and ambient densities, reaching a maximum injection pressure of 300 MPa. A prediction model for spray tip penetration under non-evaporating conditions was developed based on momentum conservation and a phenomenological model. The main conclusions are as follows:

(1) The injection rate exhibits three main stages, characterized by rapid rise, stable duration, and rapid decrease. Higher injection pressures result in faster changes in the initial and final stages.

(2) Initially, the spray tip penetration of the target spray differs from other sprays, but it gradually converges as the spray develops. Under ultra-high injection pressure conditions, the spray boundary of larger hole-diameter injectors becomes highly unstable.

(3) Spray tip penetration increases with higher injection pressure, lower ambient density, and larger hole diameter. However, the effect of injection pressure diminishes as it exceeds 200 MPa. During the initial stage, the spray tip penetration is relatively similar

under the same ambient pressure. Larger hole-diameter injectors exhibit longer spray tip penetration due to higher mass flow and momentum.

(4) Our predicted model aligns well with experimental data under ultra-high and high injection pressure conditions. Hiroyasu's and Dent's models underestimate spray tip penetration. Increasing hole diameter and decreasing ambient pressure lead to increasing errors in our model, particularly in the initial stages, which may be influenced by experimental errors, spray angle discrepancies, and uncertainties in determining the start of injection (SOI).

**Author Contributions:** Conceptualization, C.Z. and H.L.; methodology, C.Z. and H.L.; validation, H.L.; formal analysis, C.Z.; investigation, C.Z. and H.L.; resources, F.C. and Y.J.; data curation, C.Z.; writing—original draft preparation, C.Z.; writing—review and editing, H.L., F.C. and Y.J.; visualization, C.Z. supervision, H.L.; All authors have read and agreed to the published version of the manuscript.

**Funding:** This study is financially supported by the Foundation of State Key Laboratory of Engines [No. K2022-12].

**Institutional Review Board Statement:** Not applicable.

**Informed Consent Statement:** Not applicable.

**Data Availability Statement:** Not applicable.

**Acknowledgments:** The authors would like to acknowledge the Mazda Motor Corporation for their support of the equipment apparatus.

**Conflicts of Interest:** The authors declare no conflict of interest.

## Abbreviations

| | |
|---|---|
| ASOI | After the start of injection |
| CVCC | Constant volume combustion chamber |
| D | Nozzle hole diameter |
| DBI | Diffuser background illumination |
| HSV | High-speed video |
| K | Model constant |
| $K_0$ | Model constant |
| $K_1$ | Model constant |
| $K_2$ | Model constant |
| $L_b$ | Breaking length |
| $P_{inj}$ | Injection pressure |
| $P_a$ | Ambient pressure |
| $\triangle P$ | Difference between injection and ambient pressures |
| $\rho_a$ | Ambient density |
| $\rho_l$ | Fuel density |
| $r$ | Radius position of any spray cross-section |
| $R$ | Maximum radius of the cross-section |
| $S$ | Spray tip penetration |
| SOI | Start of injection |
| $t_b$ | Breaking time |
| $\tau_{KH}$ | Breaking time by KH-RT model |
| $T_a$ | Ambient temperature |
| $U_r$ | Relative drop/gas velocity |
| $v$ | Velocity of the spray cross section |
| $v_{\max}$ | Velocity of |
| $v_m$ | Velocity at the center axis of the spray cross section |
| $v_0$ | Velocity at the nozzle |
| $v_{tip}$ | Velocity of spray tip |
| $v_{Lb}$ | Velocity at the breaking length |
| $\theta$ | Spray angle |

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
