# Peer review of "Investigations on the Diesel Spray Characteristic and Tip Penetration Model of Multi-Hole Injector with Micro-Hole under Ultra-High Injection Pressure"

_sustainability, doi:10.3390/su151411114_

Round 1

Reviewer 1 Report

The paper entitled “Investigations on the Diesel Spray Characteristic and Tip Penetration Model of Multi-Hole Injector with Micro-Hole under Ultra-High Injection Pressure” is of interest. Writing is good and structure of the paper is reasonable for a research paper except at results and discussions. In another way, section 5 which could be considered as the results of the experimental and theoretical findings is just 6 lines without comparing to previous works and discussing parts of the theoretical results that do not compatible with the experimental ones in figures 9-11.

Please consider follow comment to improve the manuscript:

1.     Please enlarge the lower part of the Figure 6 (injection rate).

2.     All equations used need to be supported by a proper reference.

3.     The manuscript suffers from discussing with previous related papers.

4.     The results part requires be expanded significantly.

Regards

Author Response

Please check the attachment of  "Responses to Reviewers'

Reviewer 2 Report

Der Authors,

The article is interesting, but I believe that the manuscript in its current version should not be accepted for publication.

To improve the quality of the article, I suggest including specific comments:

1) Introduction: The sense of referring to the cited sources [1] and [5] is not clearly defined. It would be beneficial, given the purpose and subject of the article, to justify the advisability of referring to the mentioned publications.

2) Table 1: Desantes and Payri models - What is the point of citing the same model twice? Rather, it is the model of the two authors mentioned and should be treated as one.

3) Experimental setup: Is the chamber being used a combustion chamber or a chamber for visualizing and analyzing the fuel injection process? What is the maximum range of injected fuel flow that can be analyzed?

4) Table 4: Unit error for Viscosity. Also, in density units, "3 "should be inserted as superscript - this comment applies to all manuscript content.

5) Table 4: No units for Nozzle Hole Diameter (D).

6) Theoretical Model Analysis: Explain all quantities appearing in equations, such as in equation (5) Ur; tKH. This also applies to the completion of the list of Abbreviations.

7) Line 242: “Where  tb and  n  can be calculated by Eq. 5 and Eq. 6” – Is it only about equations (5) and (6), or also about equation (7)?

8) Line 279: "According to the structure of the spray in Figure 2, the following relationship can be..." - Rather, this is referring to Figure 8.

9) Lines 318-320: Repetition of equations (6) and (7).

10) Line 321: Chapter 5 has the same title as Chapter 4. A correction should be made with titles according to the content of the chapters.

11) Line 325: „…fixed values respectively, and 10 and 11 show the comparison results…” – It should be supplemented that it is about drawings.

12) Figure 9: Didn't the simulation studies consider injection pressures of 200 MPa, only 100 and 300?

13) Line 336: It would be beneficial to add in the caption that it is about ambient gas, i.e. „Comparison between model and experimental results with different densities of ambient gas”

14) Conclusions: Motion No. (4) repeats the statement at the beginning of this chapter (Lines: 342-343):

„…prediction model of penetration distance of non-evaporating Diesel Based on  momentum conservation and phenomenological model is proposed…”

In the conclusions, it would be beneficial to relate the results obtained to the work of other authors.

15) Editorial correction of the content of the manuscript should be carried out before publication.

Best regards

Editorial correction of the content of the manuscript should be carried out before publication. It contains many errors such as:

- Table 1 – Insert spaces between the values of the quantities and their unit, e.g., instead of "10-66MPa" - 10-66 MPa; these corrections should be made throughout the manuscript.

- Table 1 – In the Dent model, a space should be inserted between 1971 and [16].

- Table 2 – remove the comma after the Mini-Sac

- Table 2 – At the Hole length unit remove the space, i.e. [mm]

Line 87 – remove the space after (Electronic Control Unit)

Lines 89-90 – Common Rail should be written in capital letters

Line 98 – Remove the space - 37.5

Table 3 – I propose to standardize the way the unit is stated throughout the manuscript, i.e., either use square brackets [] or round brackets (); in addition, the unit of Exposure should be [ms].

Line 201 – there is no space between „DPis”

Line 202 – It would be advantageous to insert commas between successive described quantities; in addition, spaces are missing between „tbis”

Line 203 – Breaking time - do you use a capital letter? If so, it should be standardized in the body of the entire manuscript.

Lines 221-222 – size r should not be inserted as a subscript; in addition, the spaces "vis" and "isr" are missing

Line 238 – space is missing between „tbis”

Line 254 – t(s+1) – should not be inserted as a superscript

Line 260 – not as indexes: s=(1,2,3,...,n); R; dr; moreover no space "dris"

Line 341 – unintelligible term: "spry"?

Line 342 – "Based" in lower case (similarly in Line 368)

Line 351 – unintelligible term: "end-stage"?

Line 395 – „Pressure” – rather in lower case (as in the line 396)

Author Response

(The authors gave the same response as above.)

Reviewer 3 Report

The diesel spray charcteristic was investigated under ultra-high injection pressure of 300MPa using shadow imaging method and a new zero-dimensional of spray tip penetration was proposed based on the experimental results and other theoretial/empirical models in this paper. It is meaningful to contruct a new model for the prediction of spray tip penetration under ultra-high injection pressure conditions. However, minor revision requires in the English language and conclusions. 

Some errors requires revision.

Author Response

(The authors gave the same response as above.)

Reviewer 4 Report

Attached please find the comments.

Author Response

(The authors gave the same response as above.)

Round 2

Reviewer 1 Report

Thanks to the authors for the improvement of the manuscript, still they did not answer one of the comments :

1.     The manuscript suffers from discussing with previous related papers.

Readers think there is no previous similar work.
Could authors explain this?

Author Response

Thank you very much for your comments. Actually, Many researchers have been contributing to the model of spray tip penetration. However, the previous paper only focused on the phenomenological model or momentum model. In this study, we used the phenomenological model as a base, and divide the spray into two stages. Meanwhile, Based on the spray tip penetration discussed in the first paragraph, the second paragraph considers the tip penetration in the second stage as an extension of the growth pattern observed in the first stage, using a momentum model for analysis. In other words, the results of the model in the first paragraph will directly affect the results of the second paragraph. At the same time, the previous papers on the tip penetration models were more suitable for low injection pressures. This article focuses on injection pressures extended to ultra-high pressures of up to 300 MPa. These are completely different from the previous papers.  To more clearly to highlight the key points of our paper, we have added the following sentences in our manuscript:
Line 74: The phenomenological model was used as a base and divide the spray into two stages. Meanwhile, based on the spray tip penetration discussed in the first paragraph, the second paragraph considers the tip penetration in the second stage as an extension of the growth pattern observed in the first stage, using a momentum model for analysis. In other words, the results of the model in the first paragraph will directly affect the results of the second paragraph. Finally, the validity of this model is then verified by comparing it with experimental results under ultra-high injection pressure conditions.

Reviewer 2 Report

Dear Authors,

In my opinion, the article, with the corrections made, can be published in the journal Sustainability.

Best regards,

Author Response

Thank you very much for your help!